# DNA Methylation Analysis Identifies Novel Epigenetic Loci in Dilated Murine Heart upon Exposure to Volume Overload

**DOI:** 10.3390/ijms24065885

**Published:** 2023-03-20

**Authors:** Xingbo Xu, Manar Elkenani, Xiaoying Tan, Jara katharina Hain, Baolong Cui, Moritz Schnelle, Gerd Hasenfuss, Karl Toischer, Belal A. Mohamed

**Affiliations:** 1Department of Cardiology and Pneumology, University Medical Center Göttingen, 37075 Göttingen, Germany; 2DZHK (German Centre for Cardiovascular Research), 37075 Göttingen, Germany; 3Department of Clinical Pathology, Faculty of Medicine, Mansoura University, Mansoura 35516, Egypt; 4Department of Nephrology and Rheumatology, University Medical Center of Göttingen, 37075 Göttingen, Germany; 5Department of Clinical Chemistry, University Medical Center Göttingen, 37075 Göttingen, Germany

**Keywords:** DNA methylation, volume overload, left ventricle dilatation, heart failure, biomarkers

## Abstract

Left ventricular (LV) dilatation, a prominent risk factor for heart failure (HF), precedes functional deterioration and is used to stratify patients at risk for arrhythmias and cardiac mortality. Aberrant DNA methylation contributes to maladaptive cardiac remodeling and HF progression following pressure overload and ischemic cardiac insults. However, no study has examined cardiac DNA methylation upon exposure to volume overload (VO) despite being relatively common among HF patients. We carried out global methylome analysis of LV harvested at a decompensated HF stage following exposure to VO induced by aortocaval shunt. VO resulted in pathological cardiac remodeling, characterized by massive LV dilatation and contractile dysfunction at 16 weeks after shunt. Although methylated DNA was not markedly altered globally, 25 differentially methylated promoter regions (DMRs) were identified in shunt vs. sham hearts (20 hypermethylated and 5 hypomethylated regions). The validated hypermethylated loci in Junctophilin-2 (*Jph2*), Signal peptidase complex subunit 3 (*Spcs3*), Vesicle-associated membrane protein-associated protein B (*Vapb*), and Inositol polyphosphate multikinase (*Ipmk*) were associated with the respective downregulated expression and were consistently observed in dilated LV early after shunt at 1 week after shunt, before functional deterioration starts to manifest. These hypermethylated loci were also detected peripherally in the blood of the shunt mice. Altogether, we have identified conserved DMRs that could be novel epigenetic biomarkers in dilated LV upon VO exposure.

## 1. Introduction

Cardiac remodeling, a precursor of clinical heart failure (HF), refers to functional and structural changes of the left ventricle (LV) in response to pathological cardiac insults, which include sustained pressure overload (PO; e.g., hypertension and aortic valve stenosis), persistent volume overload (VO; e.g., mitral and aortic valve regurgitation), and myocardial infarction (MI; e.g., coronary artery disease). Cardiac remodeling is characterized by progressive LV dilatation and deteriorated cardiac function [1].

LV dilatation is, initially, a compensatory response of the failing heart to maintain cardiac output. However, this beneficial dilatation is offset by concomitant augmented myocardial oxygen consumption and increased wall stress (according to Laplace’s law) [2], initiating a vicious cycle that contributes to progressive dilatation and ultimately resulting in functional deterioration and overt HF. Therefore, dilated LV was identified to be a risk factor for HF development in the Framingham heart study [3]. Consistently, in a spontaneously hypertensive HF rat model, overt systolic dysfunction was preceded by an increased chamber volume [4]. Moreover, LV dilatation is the predominant measure used to stratify patients’ risk of ventricular arrhythmias and sudden cardiac death in HF cases [5,6], and VO seems to be a major determinant of LV dilatation in these patients. In addition, despite apparent clinical stability upon optimal drug therapy, a significant residual risk of clinical deterioration and cardiac death in HF patients remains high [7]. Therefore, it is of utmost importance to identify patients at risk for worsening HF, which may provide an opportunity to avoid the transition to a clinical event. Since LV dilatation precedes functional deterioration in HF patients [8], the identification of genetic determinants of LV dilatation can be used to identify patients at risk for adverse outcomes and intervene before LV dysfunction develops.

There is increasing amount evidence suggesting that the initiation and progression of LV remodeling are associated with altered gene expression and transcriptional reprogramming [9]. As it has an essential role in controlling gene expression, it is not surprising that perturbed epigenetics could contribute to cardiac remodeling and LV dilatation [10].

Epigenetics are heritable changes that modulate gene expression without altering genomic sequences [11]. DNA methylation, an epigenetic mechanism, involves methylation of cytosines mainly in the context of cytosine–guanine dinucleotides (CpG) to 5-methylcytosine (5mC) and is crucial for gene expression [12]. Cytosine is methylated by DNA methyltransferases (DNMT1, DNMT3a, DNMT3b) and demethylated by ten-eleven translocation methylcytosine dioxygenases (TET1-3) [13,14,15,16]. In the promoters, DNA methylation usually exerts a suppressive effect through limiting the binding of transcription factors and/or inducing chromatin condensation [17,18,19].

Altered DNA methylation in the human heart contributes to HF progression [20,21,22,23,24,25,26]. However, clinical studies share common limitations (limited availability of myocardial biopsies; heterogeneity of study populations with different age, sex, and comorbidities; scarcity of disease-free control samples). DNA methylation has therefore been extensively investigated in experimental HF induced by PO and MI [27,28,29,30,31]. However, its relevance for pathophysiology of VO-induced HF is unknown.

Since molecular responses and cardiac remodeling following VO are distinct from those elicited by PO and MI [32,33,34,35], we therefore have undertaken a genome-wide approach to analyzing cardiac DNA methylation in the failing dilated murine hearts upon exposure to VO induced by aortocaval shunt. Although VO did not markedly alter the global DNA methylation ratio, 25 differentially methylated regions (DMRs) were detected in the shunt hearts. Among them, hypermethylated loci in *Jph2*, *Spcs3*, *Vapb*, and *Ipmk* promoters were also detected in dilated LV early after shunt and were consistently detected in the blood of shunt-operated mice, suggesting them to be novel biomarkers in VO-induced LV dilatation and HF.

## 2. Results

### 2.1. LV Dilatation and Contractile Dysfunction in 16-Week-VO-Exposed Mice

WT mice were exposed to VO via aortocaval shunt, and hearts were harvested at 16 weeks after surgery (Figure 1A). The heart rate was not different between the sham and shunt mice (Figure 1B). As expected, shunt mice exposed to long-term VO exhibited maladaptive cardiac remodeling, characterized by marked left ventricle (LV) dilatation, diminished systolic function, and increased cardiac mortality (Figure 1C–E, Appendix A). Morphometric analyses showed an increased LV weight-to-tibia length, indicating cardiac hypertrophy, and increased lung weight-to-tibia length, suggesting pulmonary congestion in the shunt mice (Figure 1F,G). Cardiomyocyte hypertrophy, but no marked fibrosis, together with re-expression of fetal genes, *Nppa* and *Nppb*, were evident in shunt hearts (Figure 1H,I), which was in agreement with our previous report [36].

### 2.2. Landscape of DNA Methylation in Volume-Overloaded Dilated LV

To determine the global methylation pattern of VO-induced HF, we performed RRBS, which identifies DNA methylation across the whole genome. Globally, an average of 1.9xE8 of CpGs were detected genome-wide; however, cytosine methylation patterns in the CpG context showed no significant difference (66.15 ± 1.8%% in shunt vs. 64.92 ± 3.2% in sham) (Appendix A). Next, we focused on the DMRs in gene promoters. In total, 25 gene promoter regions (including 13 predicted genes) were differentially methylated. Among them, 20 genes were hypermethylated and 5 genes were hypomethylated (Figure 2).

### 2.3. Validation of Promoter Methylation

To validate the promoter methylation, we first excluded 13 uncharacterized genes based on a literature search and then completed a preselection from 12 known genes using an adjusted *p* value ≤ 0.05. A total of 8 genes were identified. Next, we created a serial dilution of the DNA promoter regions amplified using qRT-PCR for a primer amplification efficiency test. Out of 8 preselected genes, the MeDIP primers for 4 genes, namely Signal peptidase complex subunit 3 (*Spcs3*), Vesicle-associated membrane protein-associated protein B (*Vapb*), Junctophilin-2 (*Jph2*), and Inositol polyphosphate multikinase (*Ipmk*), achieved an amplification efficiency >99%, but not for the NLR family, pyrin domain containing 5 (*Nlrp5*), Acetylserotonin O-methyltransferase (*Asmt*), CDK5 regulatory subunit-associated protein 2 (*Cdk5rap2*), and Erythroid differentiation regulator 1 (*Erdr1*) (Appendix A). Additionally, we designed bisulfite sequencing primers for experimental validation of the 4 candidate genes (Appendix A). According to bisulfite sequencing and MeDIP-qPCR analysis, *Spcs3*, *Vapb*, *Jph2*, and *Ipmk* consistently showed increased promoter methylation levels in shunt vs. sham hearts (Figure 3).

### 2.4. Conserved Methylation Pattern in Circulating DNA

The differential methylation analyses indicated interesting new loci potentially involved in the pathogenesis of HF [22]. To search for novel peripheral biomarkers, we tested the promoter methylation patterns of these 4 candidate genes in the mouse blood using MeDIP-qPCR assay. Interestingly, the circulating DNA from shunt mice consistently showed conserved hypermethylation loci in all the tested genes (Figure 4).

### 2.5. Association of Loci-Specific Differential Methylation with the Corresponding Gene Expression

Promoter methylation mostly leads to gene silencing [17,18,19]. To identify potential associations between methylation and transcriptional activity in the VO-dilated LV, the expression of the 4 differentially methylated genes following shunt was analyzed with qPCR. Hypermethylation of *Spcs3*, *Jph2*, and *Vapb* was associated with their downregulation, indicating a direct gene silencing effect. No significant difference in the *Ipmk* expression was observed although it followed the same trend (Figure 5).

### 2.6. Similar Methylation Changes Occurred in Dilated LV Early after Shunt before Functional Deterioration

We next sought to determine if these promoter hypermethylation changes occur early after shunt before development of maladaptive remodeling. As expected, short-term VO (i.e., 1 week after shunt) induced adaptive remodeling as evidenced by LV dilatation but with improved contractility that was associated with eccentric hypertrophy with no lung congestion (Figure 6A–D). We used the same MeDIP-qPCR setup and analyzed the samples from 1-week-shunt mouse hearts. We observed, both in the heart and in the blood, a similar pattern of promoter hypermethylation. Notably, all 4 candidate genes consistently showed increased promoter methylation, albeit to a lesser extent (Figure 6E,F).

On the expression level, we detected early downregulation of *Spcs3* and *Jph2* transcripts when comparing 1-week-shunt vs. sham hearts. However, the expression of *Vapb* and *Ipmk* was not different between the groups (Figure 7A). We next overlapped the mRNA expression profile of the 25 candidate genes with our transcriptome dataset, which was previously performed with Affymetrix GeneChip microarrays analysis on the same animal model [35]. We used a cutoff *p*-value < 0.05 and log_2_FC > 1/log_2_FC < −1 to indicate the differentially expressed genes (Figure 7B). Interestingly, 3 transcripts (*Vapb*, *Jph2*, and *Ipmk*) were overlapped with the list and showed a decreased expression (Figure 7C,D).

### 2.7. No Major Changes in the Expression Levels of the DNA Methylation-Modifying Enzymes in Volume-Overloaded Dilated Remodeled LV

We next measured the mRNA levels of the DNA methylating enzymes, *Dnmts*, and DNA methylcytosine dioxygenases, *Tets*, using real-time qPCR. While 1-week VO did not alter the expression of these enzymes, long-term VO resulted in marked *Tet1* upregulation but mild *Tet3* downregulation compared to sham (Figure 8).

## 3. Discussion

The present study illustrated the DNA methylation landscape of the failing murine hearts following exposure to VO. Globally, no significant difference in methylation extent was observed between the shunt and sham hearts. However, 25 loci were found to be differentially methylated at their promoter regions. Among the top DMRs, we identified *Spcs3*, *Vapb*, *Jph2*, and *Ipmk* as novel biomarkers in dilated LV at an advanced HF stage upon exposure to pathological VO.

Cytosine DNA methylation is widely described as a transcriptional suppressive mark with the capacity to silence gene expression through preventing transcription factor binding, recruiting transcription repressors, or inducing chromatin condensation [17,18,19]. Our results consistently showed that the hypermethylated promoters of *Jph2*, *Vapb*, and *Spcs3* genes in remodeled failing hearts upon exposure to long-term VO were associated with the respective downregulation of their expression. These hypermethylated loci were consistently observed following short-term VO, with mild downregulation of *Spcs5* and *Jph2*, but no altered *Vapb* and *Ipmk* expression. Therefore, shunt induction appears to first rapidly alter promoter DNA methylation of these candidate genes in response to hemodynamic VO stress before subsequently modulating changes in their transcript levels, indicating a potential causal role of altered DNA methylation in these loci for the changes in their expression levels.

JPH2 is an essential component of excitation–contraction coupling [37,38]. Cardiac *Jph2* overexpression mice were found to have attenuated HF progression after cardiac stress [39]. *Jph2* downregulation has been found in a variety of HF patients and animal models [40,41,42,43] and is considered to be an early maladaptive molecular change occurring during transition to HF [43,44]. *Jph2* deficiency results in acute HF [43,45], and *Jph2* mutations induce hypertrophic cardiomyopathies and arrhythmias [46,47]. Here, we demonstrated an early *Jph2* promoter hypermethylation in VO-exposed hearts, which occurs before functional deterioration and is associated with downregulated cardiac *Jph2* expression, suggesting a plausible role for hypermethylated *Jph2* promoter in the progression of pathological remodeling and transition to HF upon VO exposure. IPMK is a pleiotropic protein with different physiological functions [48]. Among them, IPMK acts as Phosphoinositide 3-kinase that contributes to the activation of Akt signaling pathway [49,50]. We have previously reported that the transition from early adaptive response to maladaptive remodeling following VO exposure is associated with decreased Akt activity, with cardiac function being markedly deteriorated in Akt-knockout shunt animals [35]. Therefore, decreased *Ipmk* expression via increased promoter methylation could contribute to HF transition following VO exposure through hampering Akt activity. VAPB is an integral endoplasmic reticulum (ER) protein involved in Ca^2+^ delivery from the ER Ca^2+^ store, and its altered expression is associated with perturbed intracellular Ca^2+^ signaling [51]. Furthermore, VAPB is an essential regulator of cardiac pacemaker channels, and its deletion was found to result in bradycardia consistent with reduced excitability [52]. SPCS3 belongs to the signal peptidase complex, which cleaves signal peptides from ER-targeted faulty proteins and thus maintains a healthy membrane proteome [53]. *Spcs3* was reported to be differentially expressed in ischemic human and murine cardiomyopathy and in pressure-overloaded hearts [35,54,55]. Future loss- and gain-of-function studies are required to determine whether these identified candidates have a detrimental effect on the pathological remodeling and HF progression upon VO exposure or whether they are indeed protective.

Clinically, pathological VO, commonly observed in patients with aortic or mitral valve regurgitation and dilated cardiomyopathy, results in increased LV end diastolic volume, which seems to be a major determinant of LV dilatation. In these patients, LV dilatation remains clinically stable for months or years, but suddenly, progressive cardiac dilatation can occur, leading to HF and premature death [56,57]. Stratification of patients at-risk for decompensation is an important clinical task. Circulating methylated DNA (cmDNA) refers to DNA that has been methylated and is present in the bloodstream of an individual. This form of DNA is thought to be derived from cell-free DNA that is released from dying or damaged cells into the bloodstream [58,59,60]. Due to its stability and persistence in the circulation, cmDNA has garnered attention as a potential biomarker for early diagnosis, disease prognosis, and treatment monitoring for a variety of diseases, including cancer, cardiovascular disease, and neurodegenerative disorders [22,61,62,63,64,65]. Here, we detected conserved DMRs at the *Spcs3*, *Vapb*, *Jph2*, and *Ipmk* promoters locally in VO-dilated hearts and peripherally in the blood of shunt mice, suggesting a potentially conserved regulation of these methylation sites and further supporting their use as a novel peripheral biomarker for LV dilatation upon pathological VO exposure. However, further investigations in human samples of patients with dilated cardiomyopathy or valve regurgitation are warranted.

DNA methylation level is dynamically maintained by methylating enzymes, DNMTs and DNA methylcytosine dioxygenases, TETs [13,14,15,16]. Although long-term VO significantly increased expression of *Tet1*, global cytosine methylation levels were not markedly affected after shunt, a result, on the one side, probably attributable to a compensation by *Tet3* downregulation, as reported before [66], and on the other side, suggests a DNA demethylation-independent function of TETs enzymes, as previously indicated [67,68].

Despite several clinical studies being conducted in this area [20,21,22,23,24,25,26], the molecular mechanisms by which altered DNA methylation contributes to the progression of cardiomyopathy in HF patients remain unclear because of the genetic, environmental, and disease heterogeneity of human samples. Experimentally, transaortic constriction-induced PO and ligation of the coronary artery-triggered MI are widely used preclinical models that recapitulate HF syndrome. However, experimental VO-induced HF is far less studied despite being relatively common among HF patients due to valve regurgitation [69]. Induction of VO with aortocaval shunt mimics the increased hemodynamic preload observed in human diseases, irrespective of etiology, resulting in eccentric ventricular hypertrophy and ultimately to maladaptive remodeling and HF progression [35,36]. Therefore, a shunt model is recommended for studying VO-induced HF progression [70].

### Study Limitations

Whole LV tissues were used, and therefore the differential DNA methylation observed in the shunt hearts might have been derived from cardiac myocytes, noncardiomyocytes (e.g., fibroblasts, endothelial cells), or infiltrating inflammatory cells. Cell type-specific methylation analysis of the identified candidates will provide mechanistic insights into the cell types involved. Moreover, only young healthy WT mice were used, which is not the representative nature of HF patients, who are usually old-aged with several comorbidities. Moreover, we used only females and therefore could not determine the degree to which these findings may apply to male mice. Nonetheless, our work is the first to decipher the methylome landscape in remodeled failing hearts following exposure to VO.

## 4. Materials and Methods

### 4.1. Aortocaval Shunt

Shunt was completed as described previously [35,36]. Briefly, 8–12-week-old C57bl6/N wild-type (WT) mice (Charles River Laboratories, Sulzfeld, Germany) were anaesthetized (1–2% isoflurane), and the abdominal infrarenal aorta and inferior vena cava were exposed via midline laparotomy. A 23-gauge needle was inserted into the aorta and was pushed through at a 45-degree angle to the inferior vena cava, creating a shunt between both vessels. The aortic puncture was then closed using cyanoacrylate (Pattex, Düsseldorf, Germany). Sham animals underwent the same procedure except for the creation of the shunt and served as a control. In total, 49 mice were operated on; five of them died perioperatively (within the first 24 h post-operation) and were therefore were from our study.

### 4.2. Genomic DNA Isolation

Total genomic DNA extracted from mouse LV was prepared with DNeasy Blood & Tissue Kits (Qiagen, Hilden, Germany). The cfDNA was isolated from 250 μL of peripheral blood using a QIAamp Circulating Nucleic Acid Kit (Qiagen N.V., Hilden, Germany) following the manufacturer’s instruction. The cfDNA was eluted with 40 μL of TE buffer.

### 4.3. Reduced Representation Bisulfite Sequencing (RRBS)

For RRBS [71] of the purified mouse LV genomic DNA, library preparation with bisulfite conversion was performed with the Ovation Ultralow Methy-Seq (NuGen/Tecan Genomics, San Carlos, CA, USA) after spiking with Phi-X lambda DNA to 0.5% total mass. Libraries were sequenced on the HiSeq2000.

### 4.4. Data Processing and Mapping to the Mouse Genome

The raw sequencing data were assessed for quality using FastQC (version v0.11.7) with all samples, which are characterized by high quality base calls (Phred score > 28 across all bases). The raw sequence data were processed using TrimGalore (version 0.4.4_dev) to trim poor quality bases at the ends of reads with a quality score threshold of 20 and an error rate of 0.2. Reads with fewer than 20 base pairs after trimming were removed from further analysis [72]. The trimmed reads were then aligned to the mm10 (GRCm38) mouse genome using Bismark v. 0.19.0 [73] with the following parameters: –pbat –bowtie2 -q –score-min L,0, −0.6. The total number of aligned reads and cytosines can be found in Appendix A. Methylation levels were quantified using SeqMonk software v1.45.0 by following the pipeline described previously [74]. GRCm38 *Mus musculus* genome annotation file was adopted to annotate DMR to genes, and a gene promoter was defined as a region spanning −10 kb and +1kb around the transcription start site (TSS).

### 4.5. Methylated DNA Immunoprecipitation (MeDIP)-Quantitative Real-Time Polymerase Chain Reaction (qRT-PCR)

Loci-specific DMRs were validated using qRT-PCR. Briefly, sonicated DNA isolated from LV and blood samples was subjected to a 5-mC-specific enrichment with a Methylamp™ Methylated DNA Capture kit (Epigentek, Brooklyn, NY, USA). The processed DNA samples were then denatured, immunoprecipitated, and analyzed with qPCR using loci-specific primers (Appendix A).

### 4.6. Bisulfite Sequencing

The purified mouse heart genomic DNA was bisulfite-converted using EZ DNA Methylation (Zymo Research, Irvine, CA, USA) in accordance with the manufacture’s protocol. To amplify the bisulfite-converted DNA, a touchdown PCR program was applied with AmpliTaq Gold™ 360 Master Mix (Thermo Fisher Scientific, Bonn, Germany). The first round of PCR was performed with the annealing temperature at 60–55 °C (reduce 1 °C after each cycle). The second round of PCR was performed with a fixed annealing temperature of 55 °C. The PCR primer sequences are listed in the Appendix A. The PCR products were purified using the QIAEX II Gel Extraction Kit (Qiagen), cloned into the pGEM-T Vector (Promega, Fitchburg, WI, USA) and transformed into Top 10 Competent E. Coli Cells (Thermo Fisher Scientific). The plasmid DNA was then purified with DNA Plasmid Miniprep Kit (Qiagen), and five individual colonies for each animal were analyzed with Sanger sequencing (Seqlab, Göttingen, Germany). The methylation status was determined using a publicly available online analysis tool BISMA (http://services.ibc.uni-stuttgart.de, accessed on 16 October 2021).

### 4.7. Microarray Data Resources

The gene expression microarray dataset was downloaded from the ArrayExpress database (accession no. E-MEXP-2498) [35].

### 4.8. Quantitative RT-PCR

The DNA-free RNAs were isolated from the LV of the same mice analyzed for DNA methylation by using a RNeasy kit and the RNase-free DNAse Set (Qiagen). RNA concentration was measured with NANO 2000 (Thermo Fisher Scientific) and used for cDNA synthesis with the iScript cDNA synthesis kit (Bio-Rad Laboratories, München, Germany). Real-time PCR was performed using specific primers (Appendix A) and SYBR green fluorescent dye and calculated with the delta–delta Ct method using a Bio-Rad iQ-Cycler (Bio-Rad Laboratories).

### 4.9. Echocardiography

The mice were anesthetized with isoflurane (4–5% for induction and 1–2% for maintenance, and echocardiography was performed using Vevo2100 Imaging Software 3.1.0 (Visual Sonics, Toronto, ON, Canada). The body temperature was maintained at 38–38.5 °C using a heating pad, and heart rates were kept at 400–600 bpm. Electrocardiogram monitoring was performed using hind limb electrodes. The LV geometry and systolic function were assessed using standard 2D parasternal short-axis views in accordance with recommendations where available [75].

### 4.10. Histology

Cardiac tissues were fixed in 4% formalin, embedded in paraffin sections (6 μm), and stained with either fluorescein-conjugated wheat germ agglutinin (WGA-Alexa Fluor 594, Invitrogen, Carlsbad, CA, USA) for cross-sectional area assessment or picrosirius red (Abcam, Cambridge, MA, USA) for fibrosis.

### 4.11. Statistical Analysis

Statistical analyses were carried out using Prism software version 8.01 (GraphPad Software, Inc., La Jolla, CA, USA) with two-tailed unpaired Student’s *t*-test or one-way ANOVA with Bonferroni post-test correction where appropriate. Data are expressed as mean ± SEM. Differences were considered significant if *p* < 0.05.

## 5. Conclusions

VO alters gene-specific, but not global, DNA methylation levels in the VO-exposed dilated decompensated failing heart. The DMRs in *Jph2*, *Spcs3*, *Vapb,* and *Ipmk* promoters were conserved between the heart and blood in shunt-operated mice, suggesting them to be novel biomarkers in VO-induced cardiac remodeling and HF. The reported correlation between the differential methylation of *Jph2*, *Impk*, *Vapb*, and *Spcs3* and their transcript levels imply possible associations with HF progression. Our results may foster future investigations into the functional relevance of these methylation-sensitive candidates on the protein level to determine their precise mechanistic role in HF. Overall, our study should contribute to understanding the pathophysiology of myocardial remodeling and HF following exposure to pathological VO, a clinical condition resistant to standard therapeutic strategies for HF.

## Figures and Tables

**Figure 1 ijms-24-05885-f001:**
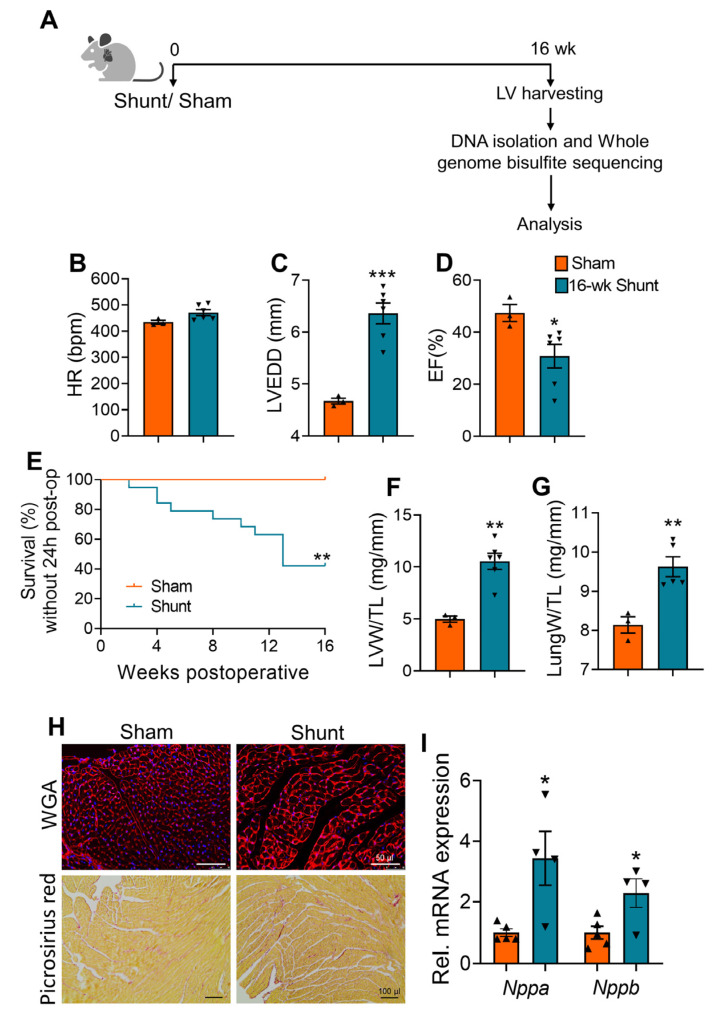
Long-term VO (16 weeks post-shunt) resulted in pathological cardiac remodeling and HF progression. (**A**) Workflow for DNA methylation analysis in VO mice. (**B**–**D**) Echocardiographic parameters: HR, heart rate (**B**); LVEDD, LV end-diastolic diameter (**C**); EF, ejection fraction (**D**). (**E**) Kaplan–Meier survival analysis, log-rank test, *n* = 9 sham and 19 shunt mice. (**F**,**G**) Morphometric parameters: LVW/TL, LV weight-to-tibia length (**F**); LungW/TL, lung weight-to-tibia length (**G**). (**H**) Representative images of WGA- (upper panels) and picrosirius red- (lower panels) stained myocardial cross-sections. (**I**) Quantitative real time PCR analysis for the expression of fetal cardiac stress genes. Values are mean ± SEM, *n* = 3–6 hearts/group. * *p* < 0.05, ** *p* < 0.01, *** *p* < 0.001 vs. sham, two-tailed unpaired Student’s *t*-test.

**Figure 2 ijms-24-05885-f002:**
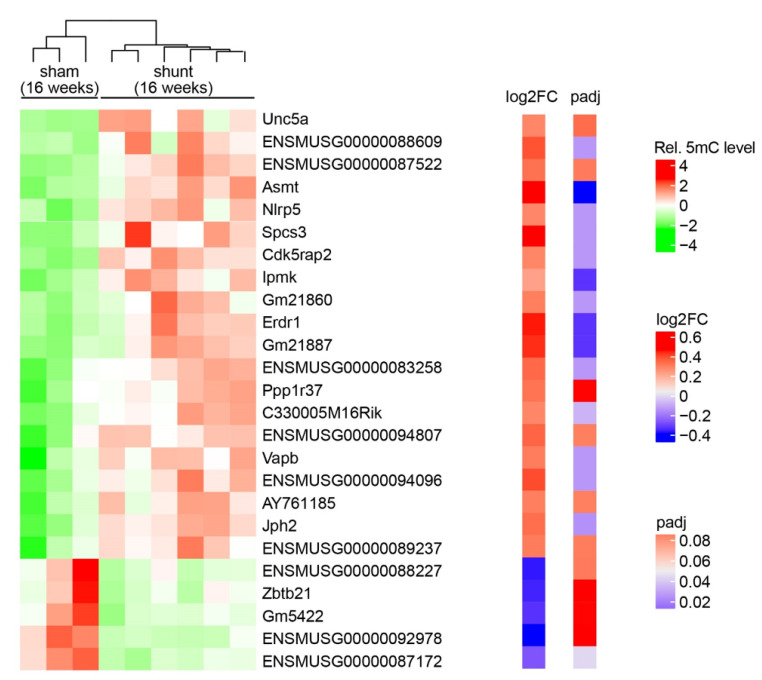
Differential promoter DNA methylation identified by RRBS. Heatmap showing the differential DNA methylation level for the candidate genes from 3 sham and 6 shunt mouse hearts. Log2FC shows the log2 value of fold change between the mean value of the sham and the mean value of the shunt. P_adj_ shows the adjusted *p*-value between the sham and shunt hearts.

**Figure 3 ijms-24-05885-f003:**
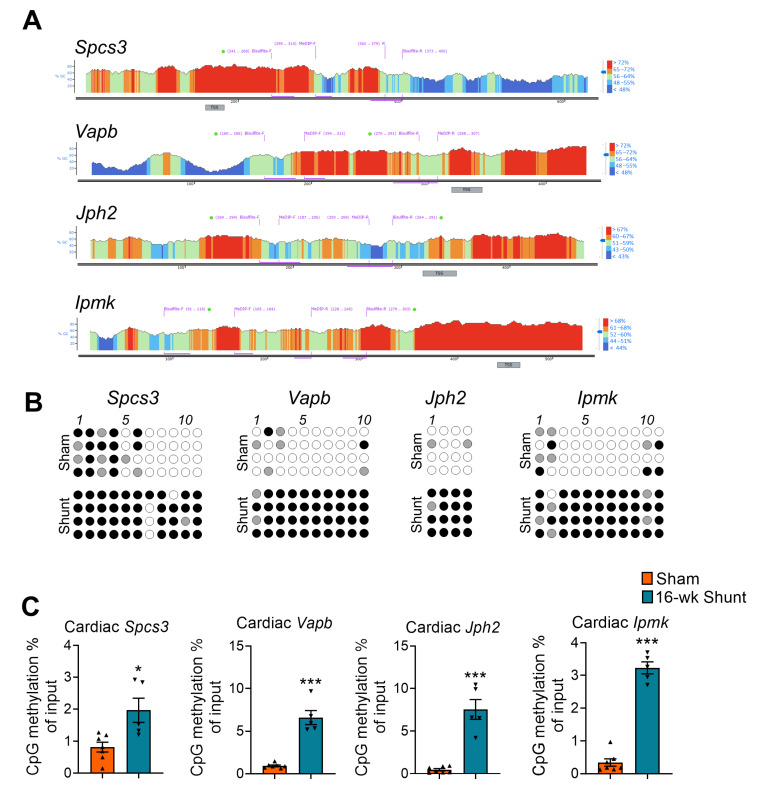
Gene-specific promoter DNA methylation assessment with bisulfite sequencing and MeDIP-qPCR. (**A**) Histogram showing the CpGs content at the gene promoter region. The locations for bisulfite sequencing and MeDIP-qPCR primers are indicated on top of the histogram. (**B**) Bisulfite sequencing analysis showing the methylation status of CpGs at the selected candidate genes. Every panel shows the sequencing results derived from four different animals. Closed circles indicate methylated, open circles indicate unmethylated, and gray circles indicate undetermined CpGs. (**C**) MeDIP-qPCR results showing the promoter methylation level for the four selected genes. Values are mean ± SEM, *n* = 4–7 hearts/group. * *p* < 0.05, *** *p* < 0.001 vs. sham, two-tailed unpaired Student’s *t*-test.

**Figure 4 ijms-24-05885-f004:**
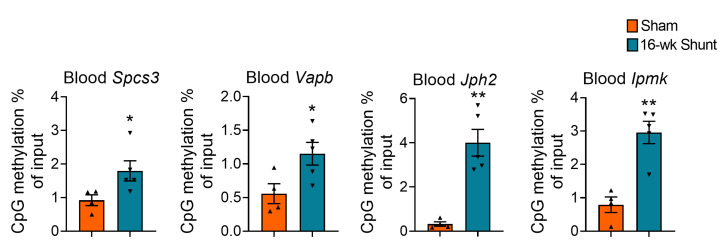
Promoter DNA methylation in the blood of shunt mice. MeDIP-qPCR results showing the promoter methylation level for the four selected genes from the DNA isolated from 16-week-shunt mice blood. Values are mean ± SEM, *n* = 4–5 mice/group. * *p* < 0.05, ** *p* < 0.01 vs. sham, two-tailed unpaired Student’s *t*-test.

**Figure 5 ijms-24-05885-f005:**
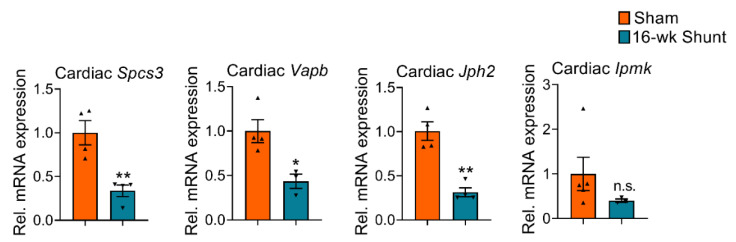
qPCR analysis showing the mRNA expression level of the four selected candidate genes in 16-week-shunt mouse hearts. Values are mean ± SEM, *n* = 3–4 hearts/group. * *p* < 0.05, ** *p* < 0.01 vs. sham, two-tailed unpaired Student’s *t*-test. n.s., not significant.

**Figure 6 ijms-24-05885-f006:**
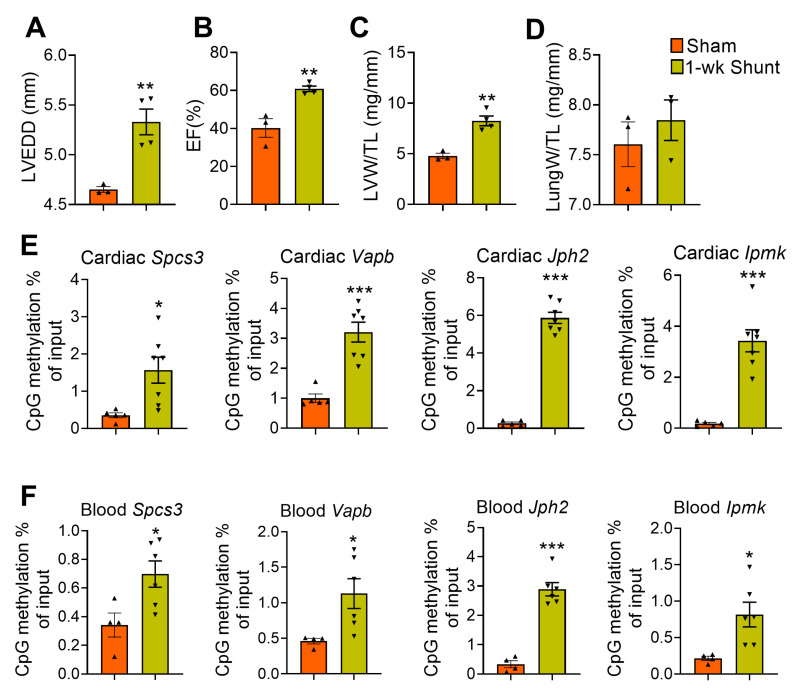
Promoter DNA methylation assessment in the hearts and blood isolated from short-term VO-exposed mice (1 week post-shunt). (**A**–**D**) Echocardiographic and morphometric parameters: LVEDD, LV end-diastolic diameter (**A**); EF, ejection fraction (**B**); LVW/TL, LV weight-to-tibia length (**C**); LungW/TL, lung weight-to-tibia length (**D**). (**E**,**F**) MeDIP-qPCR to assess DNA methylation in the hearts (**E**) and blood (**F**). Values are mean ± SEM, *n* = 3–7 mice/group. * *p* < 0.05, ** *p* < 0.01, *** *p* < 0.001 vs. sham, two-tailed unpaired Student’s *t*-test.

**Figure 7 ijms-24-05885-f007:**
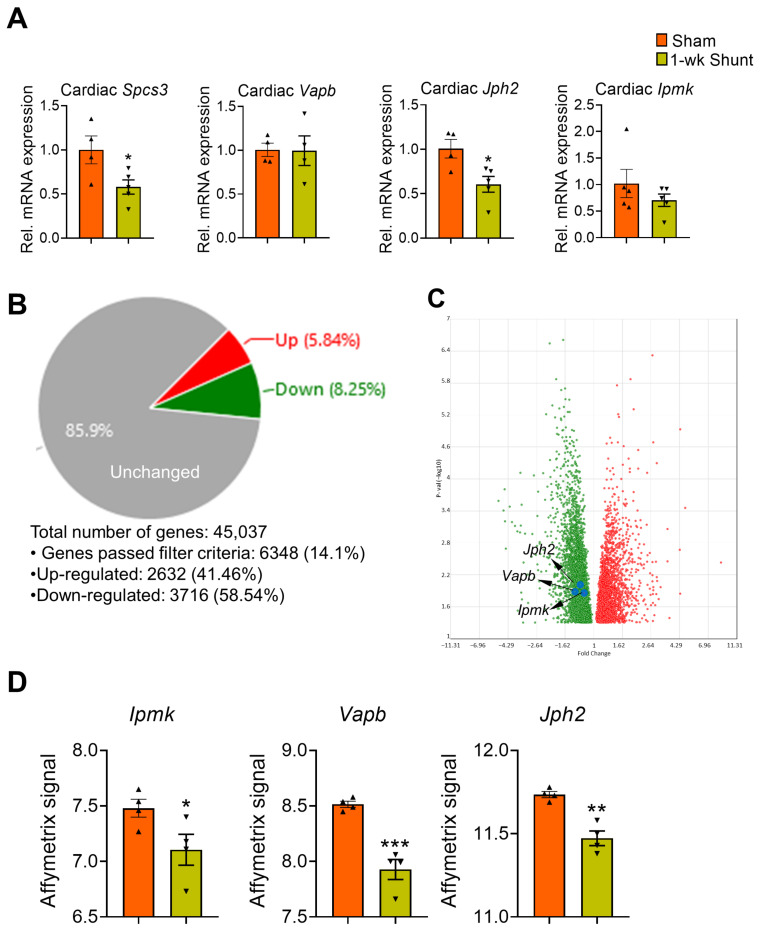
mRNA expression of the selected genes at 1 week after shunt. (**A**) qPCR analysis showing the mRNA expression level of the four selected candidate genes. (**B**) Pie chart summarizing the percentage of upregulated (5.84%), downregulated (8.25%), and unchanged (85.9%) genes in 1-week-shunt vs. sham mouse hearts. (**C**) Volcano plot shows downregulation of *Jph2*, *Vapb* and *Ipmk* in 1-week-shunt vs. sham mouse hearts. The selected candidate genes are shown with blue dots. Filter criteria: fold Change: >1 or <−1 and *p*-value < 0.05. (**D**) Barplot showing the Affymetrix signal intensity for the *Ipmk*, *Vapb*, and *Jph2* genes compared between 1-week-sham and shunt mouse hearts. Values are mean ± SEM, *n* = 4–5 hearts/group. * *p* < 0.05, ** *p* < 0.01, *** *p* < 0.001 vs. sham, two-tailed unpaired Student’s *t*-test.

**Figure 8 ijms-24-05885-f008:**
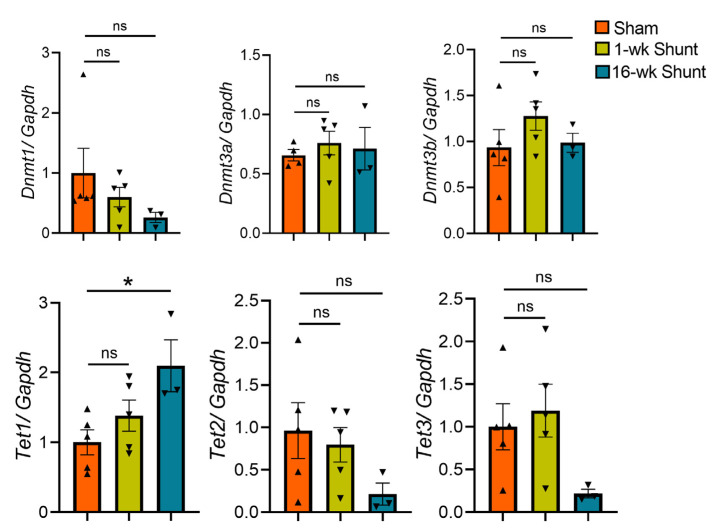
Quantitative real-time PCR of *Dnmts* and *Tets* enzymes. Values are mean ± SEM, *n* = 3–5 hearts/group. * *p* < 0.05 vs. sham, two-tailed unpaired Student’s *t*-test. ns, not significant.

## Data Availability

The datasets presented in this study are available online at https://www.ncbi.nlm.nih.gov/sra?LinkName=bioproject_sra_all&from_uid=946659 (accessed on 15 February 2023).

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
