# Peer review of "DNA Methylation Analysis Identifies Novel Epigenetic Loci in Dilated Murine Heart upon Exposure to Volume Overload"

_ijms, 2023, doi:10.3390/ijms24065885_

Round 1
Reviewer 1 Report
In this paper, the authors investigated epigenetic changes linked to VO-induced left ventricular dilatation, a risk factor of heart failure.
To this aim, the authors determined the methylation status of gene promoters in hearts harvested from mice subjected to aortocaval shunt (shunt mice) and compared it with the methylation status in mice subjected to laparotomy only (sham mice).
The authors found that the promoter region of 4 genes (Spcs3, Vapb, Jph2, Impk) is hypermethylated in shunt mice, and that this hypermethylation pattern in conserved in circulating DNA and emerges early after aortocaval shunt. For some of these genes, DNA hypermethylation is associated with decreased gene expression.
The authors conclude that the methylation status at these 4 genes can serve as potential biomarkers of cardiac remodeling induced by volume-overload.
Overall, the study is well designed. The results, supported by appropriate controls and validations, have been exposed in a clear manner in the manuscript. Here are some suggestions to further improve the manuscript.
Major
1) Materials and Methods misses some important information:
a. Total number of mice adopted in the study
b. Extraction and analysis of circulating DNA
c. Complete processing of RRBS and BS data, including FastQC and read trimming. Regarding the alignment, did the authors use the same parameters for both RRBS and BS seq data? Finally, the SeqMonk pipeline should be referenced.
d. How were promoter regions defined, and what reference gene list was adopted to annotate DMR to genes?
e. How was the relative DNA methylation level (ranging between -4 and 4) reported in Figure 2 computed?
2) In Results section, the authors should state on how many mice was their analysis carried out. For 16 weeks DNA methylation analysis, this information is implicit in the heatmap (Figure 2) and, less clearly, in some barplots (Figure 1). What about survival analysis, circulating DNA analysis and 1 week DNA methylation analysis? Was the RNA analysis performed on the same mice analyzed for DNA methylation?
3) The authors did not state on how many CpG and in which context was the global DNA methylation level reported in Section 2.2 computed. Have the authors focused only on promoter CpGs? In addition, it would be appropriate to add the standard deviation to the global DNA methylation value.
4) The authors report that 8 genes were chosen from the 12 differentially methylated known genes. How were these genes selected?
5) In Figure 3, the coordinates of MeDIP and BS primers are not readable. I suggest to zoom-in the promoter region to map the primers and the assayed CpGs in respect of the gene TSS. I also suggest to add the coordinates of the amplified regions to Supplementary Tables. Always in Figure 3, the authors should better describe the plot in panel B. I assumed that each dot represents the methylation level of an individual CpG in a mouse. If so, how were methylated and unmethylated CpG defined? Why the first 4 CpG of Spcs3 differentiate between groups only for grey circles (undetermined CpGs)?
Minor
1) The following statements should be referenced in Introduction and Discussion:
- Despite apparent clinical stability upon optimal drug therapy, a significant residual risk of clinical deterioration and cardiac death in HF patients remains high.
- Cytosine is methylated by DNA methyltransferases (DNMT1, DNMT3a, DNMT3b), and demethylated by ten-eleven translocation methylcytosine dioxygenases (TET1-3).
- Since molecular responses and cardiac remodeling following VO is distinct from that elicited by PO and MI
- Clinically, pathological VO, commonly observed in patients with aortic or mitral valve regurgitation and dilated cardiomyopathy, results in increased LV end diastolic volume, which seems to be a major determinant of LV dilatation. In these patients, LV dilatation remains clinically stable for months or years, but suddenly, progressive cardiac dilatation occurs, leading to HF and premature death.
- Circulating methylated DNA (cmDNA) refers to DNA that has been methylated and is present in the bloodstream of an individual. This form of DNA is thought to be derived from cell-free DNA that is released from dying or damaged cells into the bloodstream.
2) Data availability statement is missed.
3) Axis labeling of Supplementary Figure S1 should be added. In addition, since the curves are presented for the 4 genes that achieved 99% amplification efficiency only, consider adding the curves of the other 4 genes tested.
4) In Figure 8, consistent usage of asterisk or direct p value should be adopted for barplot 4 and 6.
Author Response
Reviewer 1
Comments and Suggestions for Authors
In this paper, the authors investigated epigenetic changes linked to VO-induced left ventricular dilatation, a risk factor of heart failure.
To this aim, the authors determined the methylation status of gene promoters in hearts harvested from mice subjected to aortocaval shunt (shunt mice) and compared it with the methylation status in mice subjected to laparotomy only (sham mice).
The authors found that the promoter region of 4 genes (Spcs3, Vapb, Jph2, Impk) is hypermethylated in shunt mice, and that this hypermethylation pattern in conserved in circulating DNA and emerges early after aortocaval shunt. For some of these genes, DNA hypermethylation is associated with decreased gene expression.
The authors conclude that the methylation status at these 4 genes can serve as potential biomarkers of cardiac remodeling induced by volume-overload.
Overall, the study is well designed. The results, supported by appropriate controls and validations, have been exposed in a clear manner in the manuscript. Here are some suggestions to further improve the manuscript.
Major
1) Materials and Methods misses some important information:
- Total number of mice adopted in the study
We have operated 49 mice. Five mice died during the first 24h after surgery and therefore were excluded. We have added one sentence in the aortocaval shunt section of the Method part to address this issue). (Please see Lines 312-313)
- Extraction and analysis of circulating DNA
The circulating DNA extraction method now has been added (please see section 4.3 in the revised manuscript). (Please see section 4.3 in the revised manuscript) (Lines 316-318)
The MeDIP-qPCR was the method which we used to analyze the methylation level of circulating DNA (please see section 4.6 in the revised manuscript). (s. Lines 337-343)
- Complete processing of RRBS and BS data, including FastQC and read trimming. Regarding the alignment, did the authors use the same parameters for both RRBS and BS seq data? Finally, the SeqMonk pipeline should be referenced.
The analysis details of RRBS processing are now included in the section of 4.5 in the reviewed manuscript. (Please see Lines 324-336)
We applied Sanger sequencing method for Bisulfite-sequencing analysis. The details were already included in the initial submission (now at section 4.7).
- How were promoter regions defined, and what reference gene list was adopted to annotate DMR to genes?
The GRCm38 Mus musculus genome annotation file was adopted to annotate DMR to genes and a gene promoter is defined as a region spanning -10kb and +1kb around the transcription start site (TSS). This information has been added to the revised version at section of 4.5. (Please see Lines 333-336)
- How was the relative DNA methylation level (ranging between -4 and 4) reported in Figure 2 computed?
In order to better visualized the base count differences between shunt and sham groups, we applied scale function which is often applied in order to visualize a dataset in a heatmap plot. The scale function centers the data by subtracting the mean from each value and divides each value by the standard deviation, and then scales it to a new range by multiplying the range by a scaling factor 2. Here is the formula which I applied for the scaling the values: scaled value = ((original value - mean) / standard deviation) *2.
2) In Results section, the authors should state on how many mice was their analysis carried out. For 16 weeks DNA methylation analysis, this information is implicit in the heatmap (Figure 2) and, less clearly, in some barplots (Figure 1). What about survival analysis, circulating DNA analysis and 1 week DNA methylation analysis? Was the RNA analysis performed on the same mice analyzed for DNA methylation?
We have added the number of mice/ group tested in each figure (please see the figure legends).
Yes, the RNA analysis was performed on the same mice analyzed for DNA methylation (please see Section 4.9, lines 363-364)
3) The authors did not state on how many CpG and in which context was the global DNA methylation level reported in Section 2.2 computed. Have the authors focused only on promoter CpGs? In addition, it would be appropriate to add the standard deviation to the global DNA methylation value.
Thank you for the suggestion. Now, we made a summary of total number of CpG identified from genome wide analysis (Table S2) and updated the global methylation percentage with the standard deviation (please see Section 2.2, lines 107 -110).
4) The authors report that 8 genes were chosen from the 12 differentially methylated known genes. How were these genes selected?
Based on the RRBS analysis, out of 25 candidate genes, we first excluded 13 uncharacterized genes based on literature search, which means 12 known genes remained. Next, we only focused on the genes with adjusted p value ≤0.05 from RRBS statistical analysis (Please see Section 2.3, lines 120-129).
5) In Figure 3, the coordinates of MeDIP and BS primers are not readable. I suggest to zoom-in the promoter region to map the primers and the assayed CpGs in respect of the gene TSS. I also suggest to add the coordinates of the amplified regions to Supplementary Tables. Always in Figure 3, the authors should better describe the plot in panel B. I assumed that each dot represents the methylation level of an individual CpG in a mouse. If so, how were methylated and unmethylated CpG defined? Why the first 4 CpG of Spcs3 differentiate between groups only for grey circles (undetermined CpGs)?
Thanks for the suggestion, a zoomed-in promoter region for each candidate gene has been added to Figure 3. Additionally, the coordinates of the amplified regions for MeDIP and BS primers have been included in Supplementary Table S3 and Table S4, respectively.
As for the panel B, the description was already included in the initial submission Figure legend 3. Bisulfite –converted DNA was amplified by BS primers for each gene and the generated amplicons were ligated into pGem-T easy vector and transformed into Top10 Competent E. Coli Cells (Thermo fisher scientific). The plasmid DNA was then purified with DNA Plasmid Miniprep Kit (Qiagen) and five individual colonies for each animal were analyzed by Sanger-sequencing (Seqlab, Göttingen, Germany). The methylation status was determined by using a publicly available online analysis tool BISMA at website of http://services.ibc.uni-stuttgart.de. Now, we have added the detailed analysis of the Bisulfite sequencing
in the revised manuscript (Please see Section 4.7, lines 355-358).
For Spcs3 gene, a clear methylation differences were only observed at the cytosine #8-#11, but not for #1-#7 between shunt and sham groups. Therefore, we purposely designed our MeDIP-qPCR primers to only cover #8- #11 in order to clearly distinguish the methylation status between shunt and sham groups.
Minor
1) The following statements should be referenced in Introduction and Discussion:
- Despite apparent clinical stability upon optimal drug therapy, a significant residual risk of clinical deterioration and cardiac death in HF patients remains high.
We have added the following reference:
Greene S.J.; Fonarow G.C.; Butler J. Risk profiles in heart failure: baseline, residual, worsening, and advanced heart failure risk. Circ Heart Fail 2020, 13, e007132.
- Cytosine is methylated by DNA methyltransferases (DNMT1, DNMT3a, DNMT3b), and demethylated by ten-eleven translocation methylcytosine dioxygenases (TET1-3).
We have added the following references:
Hermann, A.; Goyal, R.; Jeltsch, A. The Dnmt1 DNA-(cytosine-C5)-methyltransferase methylates DNA processively with high preference for hemimethylated target sites. J Biol Chem 2004, 279, 48350-9.
Okano, M.; Bell, D.W.; Haber, D.A.; Li, E. DNA methyltransferases Dnmt3a and Dnmt3b are essential for de novo methylation and mammalian development. Cell 1999, 99, 247-57.
Ito, S.; D'Alessio, A.C.; Taranova, O.V.; Hong, K.; Sowers, L.C.; Zhang, Y. Role of Tet proteins in 5mC to 5hmC conversion, ES-cell self-renewal and inner cell mass specification. Nature 2010, 466, 1129-33.
Tahiliani, M.; Koh, K.P.; Shen, Y.; Pastor, W.A.; Bandukwala, H.; Brudno, Y., Agarwal, S.; Iyer, L.M.; Liu, D.R.; Aravind, L.; Rao, A. Conversion of 5-methylcytosine to 5-hydroxymethylcytosine in mammalian DNA by MLL partner TET1. Science 2009, 324, 930-5.
- Since molecular responses and cardiac remodeling following VO is distinct from that elicited by PO and MI
We have added the following references:
Kehat, I.; Molkentin, J.D. Molecular pathways underlying cardiac remodeling during pathophysiological stimulation. Circulation 2010, 122, 2727-35.
You, J.; Wu, J.; Zhang, Q.; Ye, Y.; Wang, S.; Huang, J.; Liu, H.; Wang, X.; Zhang, W.; Bu, L.; Li, J.; Lin, L.; Ge, J.; Zou, Y. Differential cardiac hypertrophy and signaling pathways in pressure versus volume overload. Am J Physiol Heart Circ Physiol 2018, 314, H552-H562.
Pitoulis, F.G.; Terracciano, C.M.
Heart Plasticity in Response to Pressure- and Volume-Overload: A Review of Findings in Compensated and Decompensated Phenotypes. Front Physiol 2020, 11, 92.
- Clinically, pathological VO, commonly observed in patients with aortic or mitral valve regurgitation and dilated cardiomyopathy, results in increased LV end diastolic volume, which seems to be a major determinant of LV dilatation. In these patients, LV dilatation remains clinically stable for months or years, but suddenly, progressive cardiac dilatation occurs, leading to HF and premature death.
We have added the following references:
Dujardin, K.S.; Enriquez-Sarano, M.; Schaff, H.V.; Bailey, K.R.; Seward, J.B.; Tajik, A.J. Mortality and morbidity of aortic regurgitation in clinical practice. A long-term follow-up study. Circulation 1999, 99, 1851-7.
Nishimura RA, Otto CM, Bonow RO, Carabello BA, Erwin JP 3rd, Fleisher LA, Jneid H, Mack MJ, McLeod CJ, O'Gara PT, Rigolin VH, Sundt TM 3rd, Thompson A. 2017 AHA/ACC Focused Update of the 2014 AHA/ACC Guideline for the Management of Patients With Valvular Heart Disease: A Report of the American College of Cardiology/American Heart Association Task Force on Clinical Practice Guidelines. J Am Coll Cardiol 2017, 70, 252-289.
- Circulating methylated DNA (cmDNA) refers to DNA that has been methylated and is present in the bloodstream of an individual. This form of DNA is thought to be derived from cell-free DNA that is released from dying or damaged cells into the bloodstream.
We have added the following references:
Zemmour, H.; Planer, D.; Magenheim, J.; Moss, J., Neiman, D.; Gilon, D.; Korach, A.; Glaser, B.; Shemer, R.; Landesberg, G. Non-invasive detection of human cardiomyocyte death using methylation patterns of circulating DNA. Nat Commun 2018, 9, 1–9
Stroun, M.; Maurice, P.; Vasioukhin, V.; Lyautey, J.; Lederrey, C.; Lefort, F.; Rossier, A.; Chen, X.Q.; Anker, P. The origin and mechanism of circulating DNA. Ann N Y Acad Sci 2000, 906, 161-8.
Ren J, Jiang L, Liu X, Liao Y, Zhao X, Tang F, Yu H, Shao Y, Wang J, Wen L, Song L.
Heart-specific DNA methylation analysis in plasma for the investigation of myocardial damage. J Transl Med 2022, 20, 36.
2) Data availability statement is missed.
We have added the Data Availability Statement into our revised manuscript (Lines 434-435).
3) Axis labeling of Supplementary Figure S1 should be added. In addition, since the curves are presented for the 4 genes that achieved 99% amplification efficiency only, consider adding the curves of the other 4 genes tested.
We have added the axis labelling as well as added amplification curves for the other 4 genes into the Supplementary Fig. S1.
4) In Figure 8, consistent usage of asterisk or direct p value should be adopted for barplot 4 and 6.
We have removed the p value from barplot 6 to be consistent with all graphs in our manuscript.

Reviewer 2 Report
The manuscript by Xu et al identifies a series of differentially methylated promoters in a mouse model of dilated cardiomyopathy exposed to volume overload. They found that such methylated promoters are already detectable before functional deterioration in ventricular samples as well as in peripheral blood. Overall the study is very well conducted and it is relevant for the field. However, there are two major points that should be addressed.
a) The authors failed to properly demonstrate how changes in the methylation status of these selected promoters is achieved. Although the analysed the expression of distinct methylases and demethylases, they only found changes in the demethylases expression. Thus, how do they explain that the become methylated. Secondly, they only analyzed it at mRNA level, but not at protein levels. It might be important to obtain those data.
b) Within the discussion, the link between the methylated promoters (genes) and the mechanism of functional deterioration is very elusive. The authors would need to do a more elaborated integration of these results and the plausible molecular mechanisms underlying this pathology.
Author Response
Reviewer 2
Comments and Suggestions for Authors
The manuscript by Xu et al identifies a series of differentially methylated promoters in a mouse model of dilated cardiomyopathy exposed to volume overload. They found that such methylated promoters are already detectable before functional deterioration in ventricular samples as well as in peripheral blood. Overall the study is very well conducted and it is relevant for the field. However, there are two major points that should be addressed.
- a) The authors failed to properly demonstrate how changes in the methylation status of these selected promoters is achieved. Although the analysed the expression of distinct methylases and demethylases, they only found changes in the demethylases expression. Thus, how do they explain that the become methylated. Secondly, they only analyzed it at mRNA level, but not at protein levels. It might be important to obtain those data.
In our study, we did not detect any differential changes in the expression levels of methylases, namely Dnmt1, Dnmt3a and Dnmt3b, and of DNA methylcytosine dioxygenase, Tet2 in shunt vs. sham. The marked upregulation of Tet1 could be possibly compensated by Tet3 downregulation we saw in shunt vs. sham hearts. Therefore, theoretically it is not surprising that the global methylation levels were not different between shunt (0.9%) and sham (1.1%). Moreover, differential expression of Tet1 and, to less extent, Tet3, might indicate a DNA demethylation-independent function of these enzymes as previously suggested. We now included a sentence to address this issue in the Discussion (Please see page 10; lines 273-275).
- b) Within the discussion, the link between the methylated promoters (genes) and the mechanism of functional deterioration is very elusive. The authors would need to do a more elaborated integration of these results and the plausible molecular mechanisms underlying this pathology.
We totally agree with the reviewer`s comment. Although future studies are needed to examine the causal nature of the identified DMRs in HF pathogenesis following VO exposure, we think that our study has high potential to markedly contribute to improve our understanding the pathogenesis of cardiac remodeling and HF in the context of VO, a clinical condition resistant to standard therapeutic strategies. Here we identified novel hypermethylated candidates that, on one side suppress their gene expression, and on the other side occurs early after VO exposure even before functional deterioration starts to appear, indicating a possible causal role of these methylation changes in the pathogenesis of VO-induced HF. Moreover, finding a conservative methylation changes of these candidates in the blood makes our findings clinically relevant, as diagnostic and prognostic biomarkers in the context of VO following valvular insufficiency. Therefore, we believe that our study would foster future investigations into the functional relevance of these methylation-sensitive candidates, not only on RNA but also on protein levels, to determine their precise mechanistic role in HF. We have added a couple of sentences in the Conclusion section to address this issue (Please see page 13; lines 391-398).
